# Eyebirds: Enabling the Public to Recognize Water Birds at Hand

**DOI:** 10.3390/ani12213000

**Published:** 2022-11-01

**Authors:** Jiaogen Zhou, Yang Wang, Caiyun Zhang, Wenbo Wu, Yanzhu Ji, Yeai Zou

**Affiliations:** 1Jiangsu Provincial Engineering Research Center for Intelligent Monitoring and Ecological Management of Pond and Reservoir Water Environment, Huaiyin Normal University, Huaian 223300, China; 2Department of Computer Science and Technology, Tongji University, Shanghai 201804, China; 3Research Center of Information Technology, Beijing Academy of Agriculture and Forestry Sciences, Beijing 100097, China; 4Key Laboratory of Zoological Systematics and Evolution, Institute of Zoology, Chinese Academy of Sciences, Beijing 100101, China; 5Dongting Lake Station for Wetland Ecosystem Research, Institute of Subtropical Agriculture, Chinese Academy of Sciences, Changsha 410125, China

**Keywords:** bird conservation, bird identification, deep learning, deep convolution neural network, attention, image feature

## Abstract

**Simple Summary:**

Enabling the public to easily recognize water birds at hand has a positive effect on wetland bird conservation. An attention mechanism-based deep convolution neural network model (AM-CNN) is developed for water bird recognition. The model employs an effective strategy that enhances the perception of shallow image features in convolutional layers, and achieves up to 86.4% classification accuracy on our self-constructed image dataset of 548 global water bird species. The model is implemented as the mobile app of EyeBirds for smart phones. The app offers three main functions of bird image recognition, bird information and bird field survey to users. Overall, EyeBirds is useful to assist the public to easily recognize water birds and acquire bird knowledge.

**Abstract:**

Enabling the public to easily recognize water birds has a positive effect on wetland bird conservation. However, classifying water birds requires advanced ornithological knowledge, which makes it very difficult for the public to recognize water bird species in daily life. To break the knowledge barrier of water bird recognition for the public, we construct a water bird recognition system (Eyebirds) by using deep learning, which is implemented as a smartphone app. Eyebirds consists of three main modules: (1) a water bird image dataset; (2) an attention mechanism-based deep convolution neural network for water bird recognition (AM-CNN); (3) an app for smartphone users. The waterbird image dataset currently covers 48 families, 203 genera and 548 species of water birds worldwide, which is used to train our water bird recognition model. The AM-CNN model employs attention mechanism to enhance the shallow features of bird images for boosting image classification performance. Experimental results on the North American bird dataset (CUB200-2011) show that the AM-CNN model achieves an average classification accuracy of 85%. On our self-built water bird image dataset, the AM-CNN model also works well with classification accuracies of 94.0%, 93.6% and 86.4% at three levels: family, genus and species, respectively. The user-side app is a WeChat applet deployed in smartphones. With the app, users can easily recognize water birds in expeditions, camping, sightseeing, or even daily life. In summary, our system can bring not only fun, but also water bird knowledge to the public, thus inspiring their interests and further promoting their participation in bird ecological conservation.

## 1. Introduction

Water birds are an important component of natural wetland systems, and they play a key role in sustaining wetland biodiversity. Water bird conservation has become one of the important elements of biodiversity conservation in wetland ecosystems worldwide [1,2,3]. Disseminating bird knowledge to the public and promoting public participation are critical to bird conservation, especially rare and endangered birds. It is import to know or identify the taxonomic levels of birds for conservation efforts. Due to both the large number of bird species and the fact that classifying birds requires advanced ornithological knowledge, it is difficult for ordinary people to identify many bird species. This situation to a certain extent hinders the public’s enthusiasm to participate in water bird conversation. Therefore, enabling the public to recognize water birds with as little prior knowledge as possible is an effective way to promote public participation in water bird conservation. To this end, some studies have been conducted to automatically identify birds using bird sounds or images based on machine learning techniques [4,5,6].

Bird sounds are important biosignals for distinguishing different bird species. Using bird sounds as bird classification features and classifying them by machine learning methods can effectively distinguish bird species [6,7]. For example, the BirdNET model is capable of identifying 984 bird species across North America and Europe by sound [8]. Compared to the acquisition of bird sound data, the relative simplicity and ease of acquiring bird images have triggered many studies to use machine vision techniques for identifying bird images. Bird image recognition relies on the extraction of bird image features, which include color, texture, invariant geometric properties, etc. [4,5,9,10]. However, the degraded quality of image features due to the differences in image capturing pose, lighting variations and the diversity of backgrounds largely contributes to the low accuracy of traditional bird image recognition methods. In addition, the extraction of image features is traditionally achieved by constructing local feature extractors [10,11]. Whether the local feature extraction algorithm is selected appropriately or not directly affects the accuracy of image classification. Moreover, both the dependence of the selection of local image feature extraction algorithms on a priori knowledge and the non-automatic learning nature of image features further limits the application of traditional techniques for bird image recognition tasks [12].

The development of deep convolution neural networks (CNN) has made automatic learning of image features a reality [13,14]. CNN techniques have the power to automatically extract image features in an end-to-end style and can improve the network performance with extended convolution network depth by overlaying the network feature extraction components [14,15,16]. Compared with the bird classification methods described above, it is more feasible to handle bird image classification with CNN models. However, there is still one problem that affects the accuracy of CNN models in classifying bird images. High inter-class similarity of birds at the same taxonomic level, especially at the bird species level, poses a great challenge to CNN models for bird image classification [17]. To address this problem, several studies have used fine-grained image classification techniques to effectively improve the performance of bird image classification [18,19,20,21,22]. However, the fine-grained image classification task relies on fine-grained manual annotation of physiological parts such as bill, forehead, neck, torso, wings, feet and tail for each bird image. This image pre-processing is labor- and material-intensive, and it greatly limits the practical application of fine-grained image recognition techniques for bird image classification. Moreover, the current open fine-grained bird image dataset is very sparse. The publicly available fine-grained image dataset of birds (CUB200-2011) contains only 200 species of birds from North America [23]. Intuitively, local features of birds such as torso shape and feather color play an important role in distinguishing bird species, and this local information often belongs to shallow image features of birds and it also involves high-frequency signals. In fact, the successful application of attention techniques shows that strengthening local image features of interest is very effective in improving image classification [24,25,26].

Inspired by this, we propose an attention mechanism-based convolution network model (AM-CNN) for bird image classification. The model enhances local bird image features using the attention mechanism and effectively improves the accuracy of bird image classification. Our contributions consist of: (1) The construction of the first water bird image dataset, covering more than 50% of global water bird species, which is used for training the AM-CNN model; (2) The proposal of an AM-CNN model to enhance the shallow bird image features; (3) The development of an app based on WeChat applet for water bird image recognition, which is easy and convenient to deploy and use.

## 2. Materials and Methods

### 2.1. Construction of the Water Bird Image Dataset

Water bird species are numerous and widely distributed across the world. Currently, there is no open water bird image dataset available for users. Since many water bird images are scattered on various websites around the world, bird image acquisition is laborious and time-consuming. We use a web crawler technique to efficiently acquire bird image data. Chinese or English names of water bird species are used as keywords for crawling bird images. The crawled bird image dataset (Raw image sets) has many quality problems, such as low image resolution, incomplete bird image, inconsistent image matching, inconsistent image format, etc. Therefore, data cleaning is performed on raw image sets to improve the image dataset quality.

First, we remove non-bird images, caricature bird images and bird images with incomplete torsos to form a candidate image dataset of bird images. Considering the variability of bird image storage formats, all bird images in candidate image sets are uniformly converted to color images in jpg format. Second, we further refer to the Avibase website to manually check whether the mapping of all bird species with bird images or not. All bird images with inconsistent matching are removed, which ensures that each bird species file only contains the images of correctly classified birds. In addition, considering the classification performance of CNN models heavily dependent on the image number, a given bird species with less than 20 images available is discarded.

After the above operations, we constructs a professional water bird image database containing >20.5 k manually classified bird images. It covers 48 families, 203 genera and 548 species of water birds worldwide and is finally used for the later modeling of bird image identification.

### 2.2. The Eyebirds System

#### 2.2.1. Architecture

The proposed EyeBirds system is an automatic classification system for global water bird images. It consists of three main components (Figure 1): (1) a global waterbird image dataset, (2) an attention mechanism-based CNN model (AM-CNN) and (3) a user-oriented app. The water bird image dataset covers 548 water bird species worldwide, and is organized by bird species. Each bird species’ images are manually pre-processed to ensure that each bird species file only contains the images of correctly classified birds. The AM-CNN model employs an attention mechanism technology to enhance the shallow features of bird images to improve image classification performance. The app product is an intermediate component, which bridges the interaction between smartphone users and the system.

The structure of EyeBirds includes a user-side app and a server-side model. The user-side app chooses the WeChat applet as the user platform. The WeChat applet has good enough cross-platform capability with more than 1 billion users in China, and is also easily accepted by the public. The developed app based on the WeChat platform is easy to spread among public users. In the server side, we use a Flask framework to implement logic operations, model calls and other operational services. The Nginx+gunicorn technology is used to ensure that the system runs properly under concurrent scenarios of server-side tasks when multiple users are using it at the same time. By using this solution, the system server achieves the basic load balancing, high concurrency, static interception of requests and other necessary functions and ensures that the client process runs.

Overall, after a user inputs a bird image with the app of EyeBirds on the client side, the image is passed to the sever. On the server side, the logic layer implemented by Flask calls the AM-CNN model to classify the image and returns the category information of the image to the client user. The system also collects bird images uploaded by users and keeps updating the bird database with manual assistance. The classification performance of the system will be gradually improved by updating the bird image database and retraining the AM-CNN model simultaneously.

#### 2.2.2. Attention Mechanism-Based Deep Convolution Neural Network

Both the large number of bird species and the small differences in appearance among bird species pose a great challenge to the bird image recognition task in the real world. Morphological features such as bird shape, local color and detailed outline of each torso part are often important indicators for identifying bird species. Specifically, we use local color to refer to a variety of color variation, including plumage variation, bill color variation and juvenile/adult variation. Moreover, in deep convolutional neural networks, these local bird image features tend to be high-frequency signals, which are easily perceived in shallow networks but are not easily perceived in deep networks as the network depth scales. This means that enhancing bird shape and shallow image features is helpful to improve the classification performance of CNN models. Research results on attention mechanism techniques confirm that reinforcing image features in the target region brings a positive effect on model performance [24,27,28], while the success of deep dense networks also shows the positive effect of reusing shallow network features [29,30,31,32,33].

Inspired by this, we propose an attention mechanism-based deep convolution neural network model (AM-CNN).The core ideas of the model are (1) reinforcing bird shape features by using an attention mechanism technique; (2) reusing shallow convolutional image features of birds by using a spatial pyramid pooling technique. The AM-CNN model is an extension of the backbone network of Resnet18 [34]. The framework of AM-CNN includes attention layer, primary convolutional layer (PCL), convolutional block layers (Block1, Block2, Block3 and Block4), shallow network feature extraction layer (SFEL), fully connected feature layer and classification layer (Figure 2). In the model of AM-CNN, the network components of the convolutional block layers, fully connected feature layer and classification layer are consistent with the backbone network and will not be further introduced in this paper. In the later sections, we will focus on the attention layer, PCL and SFEL.

**Attention layer.** The most informative area of a bird image is the bird area, and the background region information in the image negatively affects the bird classification. Therefore, we construct an attention mechanism layer to enhance bird shape features. First, we pretrain the MMdetection network model [35] on the bird dataset of the public dataset COCO [36] to obtain the masks of bird shape. The trained MMdetection model is used to extract the mask of the water bird shapes in a given input image. The bird dataset of COCO contains fine annotation of the appearance of various types of birds, which can effectively reduce the need for shape annotations of water bird images.

Given an input color image $IMG_{i}$ with a fixed scale size of 224×224×3, the MMdetection model output the mask $M_{i}$. The mask is a black and white binary image with a scale size of 56×56. Points with a pixel value of 0 in the $M_{i}$ image correspond to the background area, and points with a pixel value of 1 correspond to the bird image area. The image $M_{i}$ is represented by the matrix *U*. Any $U\left(i,j\right)$ indicates the pixel value corresponding to the index location $\left(i,j\right)$ in matrix $U\left(1⩽i⩽56,1⩽j⩽56\right)$. The mapping of the mask $M_{i}$ is further performed to obtain the weighted feature map $A_{i}$ according to Equation (Equation 1).
(1)$$A_{i}=\left\{\begin{array}{cc} U(i,j)*\omega , & \mathrm{if}\;U(i,j)=1;\\ U(i,j)=1, & \mathrm{if}\;U(i,j)=0. \end{array}\right.$$

The $A_{i}$ serves to enhance the features of the bird body region in the bird image of $IMG_{i}$. The weight scaling of $A_{i}$ is regulated by the parameter ω. Too-large or too-small values of ω can have a negative impact on the model performance. The effect of the parameter ω on the model performance is discussed further in subsequent sections.

**Primary convolutional layer.** The primary convolutional layer (PCL) is used for shallow feature extraction of any input image $IMG_{i}$. It contains two convolutional layers: a convolutional layer of the size of 7×7×64 and a maximum pooling layer of the size of 3×3×1. The input $IMG_{i}$ image is successively processed by 7×7×64 convolutional layers and Max pooling layer, and finally, the output convolutional feature size is 56×56. The pooled output feature map is subjected to a dot product operation with the weighting feature map $A_{i}$, and then enters the convolutional block layers.

**Shallow feature extraction layer.** The shallow feature extraction layer (SFEL) is used for the acquisition of shallow convolutional features of the perceptual field at different scales in bird images. The SFEL is constructed according to the spatial pyramidal pooling method (SPP). Related studies show that SPP has the good ability to extract convolutional features of images at different scales while preserving the spatial information of the features [37,38,39,40]. In the AM-CMM model, the SFEL is connected behind the first Block1 layer in the convolutional block layers. The advantage of connecting SFEL behind the Block1 layer is that (1) the Block1 has shallow convolutional features of bird images, which retain useful classification information such as bird shape, color and line information. Extracting shallow convolutional features with SFEL can allow the AM-CMM model to pay more attention to the high-frequency signals of useful bird image features, and is beneficial to improving the classification accuracy of the model for bird images. Finally, the output of the SFEL is directly concatenated with the average pooled features and the output features of the Block4 layer, and then used for the classification process.

### 2.3. Implementation of the EyeBirds System

The structure of EyeBirds includes a user-side app and a server-side model. The server side of the system is deployed on a local server, and the relevant hardware and software configurations are shown in Table 1. The user side is developed based on a WeChat applet for smartphones with Android systems. Before developing the app, we first conducted questionnaires in public places such as parks, zoos and wetland reserves in Huai’an area to gauge the public’s demands for the app features. Overall, the top three functional requirements are bird image recognition, bird information introduction and bird field survey. Therefore, our app mainly provides these three functions (Figure 3).

The mobile app is a Chinese language version, and it is easy and convenient to operate. Under the condition of a smartphone network connection, users scan the QR code of WeChat to open the app directly. Users can obtain the classification information of the targeted birds from the system, when users upload the images of birds of interest to the system. The EyeBirds system receives and processes user task requests, and finally, returns bird information about the top 3 bird species in terms of matching confidence (Figure 3B,C). Users can also use the app’s bird field survey function to record the geographic coordinates of bird sites found, as well as the names of bird species and the number of bird species (Figure 3D).This will further enhance the user experience of the product and present bird knowledge in a pleasant way.

### 2.4. Experimental Protocol

The classification performance of the AM-CNN model is evaluated on two bird image datasets: the North American bird dataset (CUB200-2011) [23] and our self-constructed global water bird image dataset. The CUB200-2011 dataset is a benchmark bird image dataset commonly used in the deep learning field. It contains 200 bird species in North America, and each bird image contains fine annotations of more than ten bird parts. In all model training processes, bird image datasets are sampled randomly, with 80% selected as training samples, 10% selected as validation samples, and the remaining 10%—as testing samples. Model performance is characterized by a classification accuracy metric. Classification accuracy is the ratio of the number of correct predictions in the testing sample to the number of testing samples. The initial learning rate is set to 0.005, and the learning rate is reduced to the current 10% after every 7 rounds. A total of 50 rounds are trained, and a validation is performed at the end of each training session.

## 3. Results and Discussion

### 3.1. Effect of the Parameter ω on AM-CNN Performance

In the AM-CNN model, the parameter ω dominates the weighting scaling of the attention region. Too-large or too-small values of ω can have a negative impact on the model performance. In fact, the attention region is the mask region generated by the MMdetection model. The attention region is not entirely a bird region, but it also contains a certain percentage of background or error region. Therefore, the non-bird body region features are also enhanced by weighting the attention region. This indicates that the larger the ω value, the more non-bird body regions are introduced and the greater the negative impact on the model performance.

We use the North American bird dataset as a testing dataset to study the effect of the changing ω value on model performance. The experimental results show that the model classification performance has an increasing trend when the ω value increases from 1.00 to 1.20, but it decreases with the ω value over 1.20 (Table 2). In general, the best performance of the model is achieved when the ω value is 1.2. This indicates that the parameter ω taking a value of 1.20 is a relatively good balance point, where the features of the bird body region are enhanced and the effect of non-bird body regions is weakened.

### 3.2. Performance on the North American Bird Dataset

Using the bird image benchmark of CUB200-2011 as the testing dataset, we further evaluate the performance differences between the AM-CNN model and the nine existing CNN models. These nine models can be divided into two categories according to whether or not they use auxiliary annotations. First, the six models using auxiliary bird torso annotations, include Part-RCNN [18], DeepLAC [21], MG-CNN [20], PA-CNN [19], SPDA-CNN [41] and B-CNN [42]. Second, the three models without any additional annotation information contain PDFR [21], TLAN [43] and DVAN [44]. The experimental results on the CUB200-2011 dataset show that the classification performances of different CNN models differ significantly (Table 3). The average classification accuracy of the AM-CNN model is 85%, and it is better than the three CNN models of TLAN, DVAN and PDFR without applying additional annotations (classification accuracy of 77.9–82.6%). Compared with models using additional annotation information, AM-CNN outperforms DeepLAC, Part-RCNN and MG-CNN with the accuracy of 80.3–83.0%, and it is close to the models of B-CNN and SPDA-CNN with the accuracy of 85.1%. It is obvious that enhancing the shallow image features of birds effectively improves the classification accuracy of AM-CNN for bird images.

### 3.3. Performance on Our Water Bird Dataset

We further evaluated the classification accuracy of the AM-CNN model on our self-constructed global water bird image dataset. The experimental results show that there are large differences in the frequency percentage distributions of classification accuracy at the 3 taxonomic levels of family, genus and species. At the family level, the accuracy percentages falling in the less than 85% interval are 6.3%, 10.4% for 85–90%, and 83.7% for more than 90% (Figure 4A). At the genera level, the accuracy percentages falling in the less than 85% interval are 9.8%, 10.4% for 85–90% and 79.8% for more than 90% (Figure 4B). At the species level, the accuracy percentages falling in the less than 85% interval are 33.7%, 17.3% for 85–90%, and 49% for more than 90% (Figure 4C).

The above experimental results show that the classification performance of AM-CNN decreases with the hierarchical subdivision of birds. This may be related to the changes in the number of bird categories and the number of bird images as well as the similarity of birds between bird categories. In fact, from bird family to bird species, the number of bird categories increased from 48 to 548, and the average number of images per bird category decreases from 3856 to 347 (Figure 5). In addition, the similarity between birds is stronger with the hierarchical subdivision of birds. The larger number of bird categories, the smaller number of images and the higher inter-bird similarity explain well why the classification accuracy at the species level is lower than those at both the family and genus levels.

In general, AM-CNN exhibits excellent classification performance with an average classification accuracy of 94.0%, 93.6% and 86.4% at the three bird classification levels of family, genus and species, respectively. This further confirms our idea that enhancing the shallow image features of birds with the attention mechanism improves the classification performance of the model.

### 3.4. Advantages and Disadvantages of the AM-CNN Model

The great similarity in appearance among bird species poses a great challenge to bird image classification tasks. Although fine-grained image classification techniques have achieved high classification accuracy in bird image classification tasks [20,21,22], their need for additional image annotation is extremely labor-intensive and costly. In view of the important role of local image features of birds in distinguishing birds, we propose the AM-CNN model that focuses on local features of birds. The AM-CNN model uses the attention mechanism layer and shallow feature extraction layer to extract bird-shape features, and shallow features of the bird images, respectively. Compared with nine existing bird image classification models, AM-CNN performance is significantly higher than the three models (PDFR [21], TLAN [43] and DVAN [44]) without additional image annotations, and higher or close to the six models (Part-RCNN [18], DeepLAC [21], MG-CNN [20], PA-CNN [19], SPDA-CNN [41] and B-CNN [42]) with additional image annotations. This shows that our strategy of focusing on the shallow features of bird images is efficient. The mechanism behind this is that as the number of layers in a deep convolutional network increases, shallow features or high-frequency signals are often not perceived, but these shallow features play an important role in distinguishing birds. Therefore, enhancing the model’s perception of shallow features of bird images can effectively improve the classification performance. In fact, many studies have also shown that using attention mechanisms or focusing on shallow information can effectively improve model performance [27,28,31,32,33].

The performance of the AM-CNN model based on deep learning techniques is inevitably influenced by the number of bird images. In the classification tests on our self-built water bird image dataset, the AM-CNN performance decreases with the number of bird images (Figure 3 and Figure 4). In particular, the classification accuracy for some water birds with a small number of images is less than 70%. However, the AM-CNN model has been integrated into the EyeBirds system, and users can call AM-CNN through the EyeBirds app. In the future, the AM-CNN model performance will be further improved with the increase of bird images provided by users.

## 4. Conclusions

The AM-CNN model is developed to recognize 548 water bird species worldwide. The model uses an effective strategy to enhance the shallow features of bird images. Our experimental results also demonstrate that enhancing the model’s perception of shallow features of bird images effectively improves the model classification performance. The model is implemented as a mobile app for smart phones, which is free for public use. Overall, our research efforts not only provide an efficient algorithm for bird image classification, but are also beneficial to helping the public easily identify water birds and acquire bird knowledge.

In the future, we will further expand the bird image database in size and add other birds, such as forest birds, to make the system more powerful. Meanwhile, we will provide versions of the app in other languages so that the app can be available and usable beyond the Chinese world. We will also evaluate whether app usage will lead to positive attitudes of the public towards water bird and wetland conservation over time or not by means of a user tracking program.

## Figures and Tables

**Figure 1 animals-12-03000-f001:**
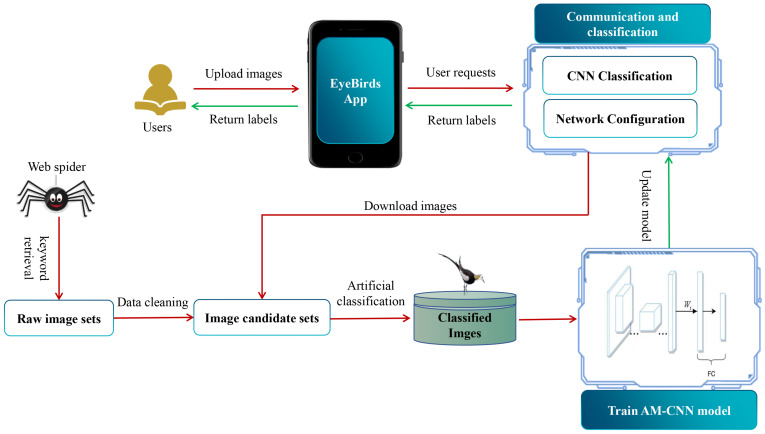
The architecture of the Eyebirds system.

**Figure 2 animals-12-03000-f002:**
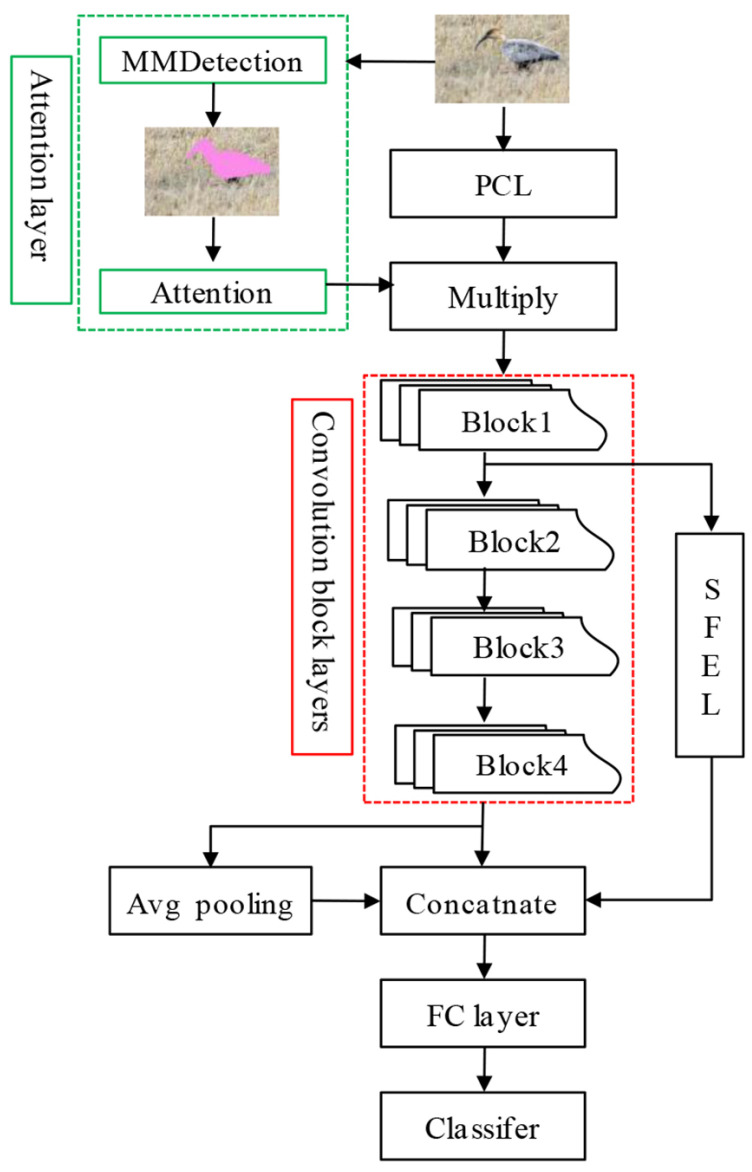
The architecture of AM-CNN model.

**Figure 3 animals-12-03000-f003:**
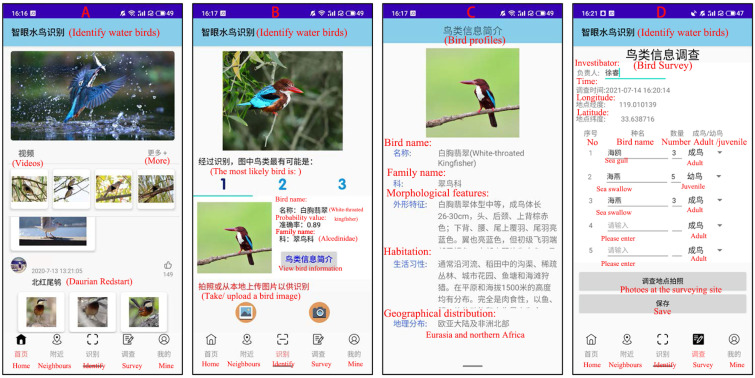
Some screenshots of the EyeBirds app interface. (**A**) App main interface. (**B**) Top 3 predicted results. (**C**) Bird information for the top 1 prediction. (**D**) Bird field survey. The English explanations of the morphological features and habits of white-throated kingfisher in (**C**) are as follows: (1) This species is medium-sized, 26-30cm in length for adults. Its feathers of head, nape and upper back are brownish-red, and lower back, and upper tail and tal feathers are bright blue. (2) It often hunts along rivers, rice ditches or water bodies in jungles and urban gardens, fishponds and beaches. It is a carnivore that feeds mainly on fish and is found on the plains and at altitudes of up to 1500 m.

**Figure 4 animals-12-03000-f004:**
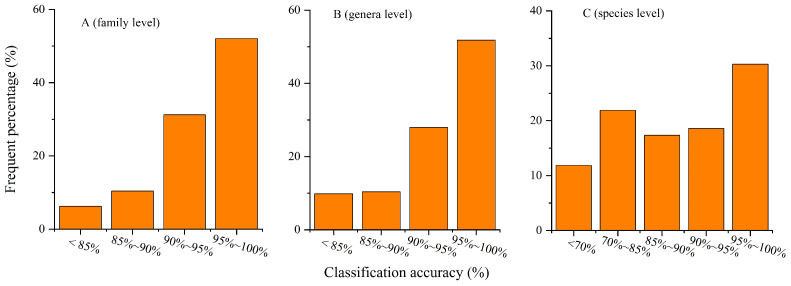
Classification accuracy at the three bird taxonomic levels of family, genus and species in our dataset. (**A**) At family level, (**B**) at genus level and (**C**) at species level.

**Figure 5 animals-12-03000-f005:**
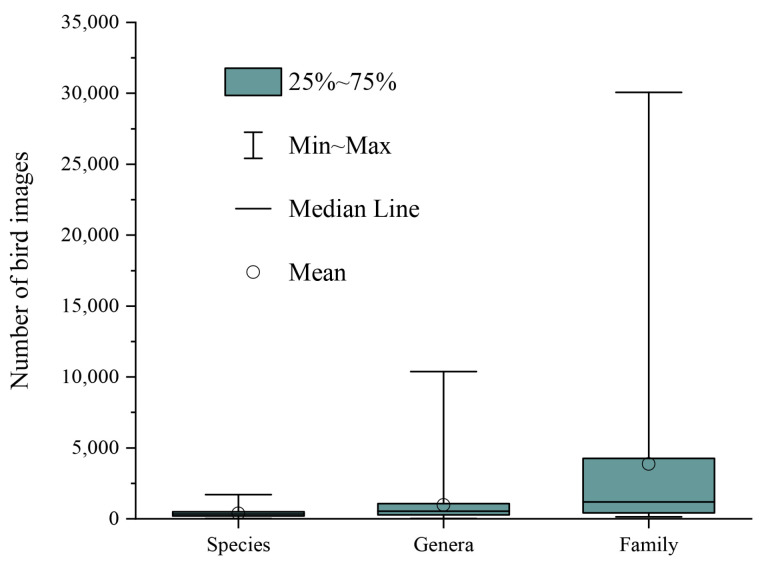
The statistics of water bird classes at the three taxonomic levels of family, genus and species in our dataset.

**Table 1 animals-12-03000-t001:** Configuration of the EyeBirds system.

No	Item	Specification
1	CPU	Intel (R) Xeon Silver, Intel (R) Xeon Silver, 2.2 GHZ processor with 40 cores
2	Hard drive and operating memory	SSD: 2T, DDR: 128 G
3	GPU	Dual channels (RTX 2080)
4	Linux operating system	CentOS 7.3
5	HTTP dynamic/static page service	Flask, Nginx + gunicom
6	Deep learning framework	Pytorch1.0

**Table 2 animals-12-03000-t002:** Effect of the parameter ω on the AM-CNN model.

No	Values	Accuracy (%)
1	ω=1.00	82.77
2	ω=1.05	82.43
3	ω=1.10	82.82
4	ω=1.15	83.60
5	ω=1.20	**84.56**
6	ω=1.40	83.43
7	ω=1.60	83.30
8	ω=1.80	83.42
9	ω=2.00	83.04
10	ω=3.00	82.36
11	ω=4.00	82.60
12	ω=5.00	82.43

**Table 3 animals-12-03000-t003:** Experimental results on the CUB200-2011 dataset.

No	Methods	Additional Annotation	Accuracy (%)
1	Part-RCNN	YES	81.6
2	DeepLAC	YES	80.3
3	MG-RCNN	YES	83.0
4	PA-CNN	YES	82.6
5	B-CNN	YES	85.1
6	SPDA-CNN	YES	85.1
7	PDFR	NO	82.6
8	DVAN	NO	79.0
9	TLAN	NO	77.9
10	**AM-CNN**	NO	**85.0**

## Data Availability

The URL for the link to the North American bird dataset is: http://www.vision.caltech.edu/datasets/cub_200_2011/, accessed on 2 February 2021.

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
