# Peer review of "Eyebirds: Enabling the Public to Recognize Water Birds at Hand"

_animals, 2022, doi:10.3390/ani12213000_

Round 1

Reviewer 1 Report

The authors present an interesting manuscript on automatic bird recognition tool. Study design and research questions are clearly described. In this sense, it is easy to understand the aim of this study. The bright side of the manuscript is that to provide some useful practical details on related topic. In this context, the study contributes to different fields. However, there are some missing points in the manuscripts. Therefore, I would like to make some suggestions to improve the quality of the paper as below:

IIntroduction

Lines 27-29: “Birds are usually classified scientifically and systematically at the levels of order, family, genus and species according to their shapes, colors and other physical characteristics.” This sentence makes no sense since all organisms have taxonomic levels of order, family, genus according to traditional taxonomy. Authors may say that “It is import to know or identify the taxonomic levels of birds for conservation efforts.”

Line 47: “Bird image recognition relies on the extraction of bird image features, which include color, texture, invariant geometric properties etc” Please add this reference here (DOI: 10.3390/ani10071207).

Lines 85-87: “Overall, our work not only is beneficial to the public to easily identify water birds and acquire bird knowledge, but also can promote public awareness and participation in bird ecological conservation” I think, this sentence belongs to Conclusion section. Please move this sentence to conclusion.

Materials and Methods

Lines 117-118: “The waterbird image dataset covers 548 species of water bird images across the world, and all bird images are manually selected and identified.” Authors should clarify how this bird species identified. I mean, did a bird expert identify the taxonomic level (species, genus, and family) of each bird? or how?

Which references were used to identify the birds’ taxonomic levels such bird of North America, Birds of The World etc.? Authors should clarify.

Line 142-143: “Morphological features such as bird shape, local color and trunk geometry are often important indicators for identifying bird species.”

What do you mean by “local color”?

Feather color variation or bill color variation?

Geographic variation in color? or sexual or juvenile/adult variation in color?

Authors should clarify.

Results and Discussion

The Discussion section should be enriched with a more theoretical interpretation and relating the present results with additional concepts. For instance, the study results can be discussed in the framework of bird recognition and bird classification in broader context.

What are the advantages and disadvantages of the application? What are the differences of the application from other similar applications? These can be explained in discussion section.

Lines 212-213: “The classification performance of the AM-CNN model is evaluated on two bird image datasets: the North American bird dataset (CUB200-2011) [21]”. Usage of North American bird dataset should be mentioned in the Methods section.

Conclusion

Line 287: The study contributes to citizen science, and this can be mentioned at the end of the Conclusion section with such a sentence or similar sentence that mention the contribution to citizen science “Our application can be used by anyone who have interest to water birds. In this context, our study contributes to volunteers for birdwatching/bird counting project and citizen science”

Author Response

Dear reviewers,

Thank you for your letter and comments concerning our manuscript entitled “Eyebirds: Enabling the Public to Recognize Water birds at Hand” (Manuscript ID: animals-1970104). Your comments have proved very helpful with regard to our revisions and improvements of our paper, as well as providing insightful guidance for our research. We have gone through your comments carefully and have made necessary corrections. We have tried our best to improve the manuscript and we have made many changes to the text according to the comments of reviewers. All changes have been highlighted in red in the revised manuscript.

We sincerely appreciate the effort and consideration of reviewers and we hope that the corrections will meet with your approval. Once again, we thank you very much for your insightful comments and suggestions.

Here, we attached a summary of our major changes in the revision.

(1) We have revised unclear statements and some inaccurate words to make the manuscript more readable and concise.

(2) Based on the reviewers' suggestions, we have revised some inaccurate or redundant statements and added some important references.

(3) Referring to the reviewer's suggestion, we have added a description of the app's operation mechanism and application cases in section 2.3.

(4) Following the reviewer's suggestion, we have added a new section 2.4 to provide additional information on the experimental configuration and the model performance evaluation metrics.

(5) In light of the reviewers' suggestions, we have added a new section 3.4 to discuss the strengths and weaknesses of the model performance.

(6) Finally, we have reorganized the conclusions by combining the reviewers' suggestions.

Our point-to-point response to the comments is as follows:

 (1) Lines 27-29: “Birds are usually classified scientifically and systematically at the levels of order, family, genus and species according to their shapes, colors and other physical characteristics.” This sentence makes no sense since all organisms have taxonomic levels of order, family, genus according to traditional taxonomy. Authors may say that “It is import to know or identify the taxonomic levels of birds for conservation efforts.”

Answer: Good suggestion. In the revised draft, we have made changes according to the suggested changes (Lines:26-27).

(2) Line 47: “Bird image recognition relies on the extraction of bird image features, which include color, texture, invariant geometric properties etc” Please add this reference here (DOI: 10.3390/ani10071207).

Answer: Good suggestion. In the revised manuscript, we have added the reference. (Lines:366-367).

(3) Lines 85-87: “Overall, our work not only is beneficial to the public to easily identify water birds and acquire bird knowledge, but also can promote public awareness and participation in bird ecological conservation” I think, this sentence belongs to Conclusion section. Please move this sentence to conclusion.

Answer: Good suggestion. In the revised manuscript, we have moved the sentence to the conclusion section and updated the contents of the conclusion.

(4) Lines 117-118: “The water bird image dataset covers 548 species of water bird images across the world, and all bird images are manually selected and identified.” Authors should clarify how this bird species identified. I mean, did a bird expert identify the taxonomic level (species, genus, and family) of each bird? or how? Which references were used to identify the birds’ taxonomic levels such bird of North America, Birds of The World etc.? Authors should clarify.

Answer: Good suggestion and thanks. there is an error in our expression of this sentence in the manuscript. In fact, what we really mean is that “The water bird image dataset covers 548 water bird species across the world, and is organized by bird species. Each bird species images are manually pre-processed as described in section 2.1 to ensure that each bird species file only contains the images of correctly classified birds”. We have made changes in the revised version (Lines:110-112).

The manual processing process is described in detail in section 2.1. In brief, we refer to the Avibase website to manually check whether the mapping of all bird species with bird images is correct or not. All bird images with inconsistent identification are removed, which ensure that each bird species file only contains the images of correctly classified birds.

(5) Line 142-143: “Morphological features such as bird shape, local color and trunk geometry are often important indicators for identifying bird species.” What do you mean by “local color”? Feather color variation or bill color variation? Geographic variation in color? or sexual or juvenile/adult variation in color? Authors should clarify.

Answer: Good suggestions. In the context, the term “local color” is used in a general rather than a specific sense in our manuscript. There are many factors that contribute to local color differences in birds, including feather color variation, bill color variation, and juvenile/adult variation, etc. These local colors belong to one of the shallow image features that we emphasize in the manuscript. Focusing on these local features is helpful to improve the performance of our model. We have modified the sentence to “Morphological features such as bird shape, local color and trunk geometry are often important indicators for identifying bird species. Specifically, we used local color to refer to a variety of color variation, including plumage variation, bill color variation, and juvenile/adult variation.” (Lines:137-139)

(6) The Discussion section should be enriched with a more theoretical interpretation and relating the present results with additional concepts. For instance, the study results can be discussed in the framework of bird recognition and bird classification in broader context. What are the advantages and disadvantages of the application? What are the differences of the application from other similar applications? These can be explained in discussion section.

Answer: Good suggestions and thanks. Following the reviewer's suggestion, we have added section 3.4 to the revised manuscript to discuss in detail the strengths and weaknesses of the model (lines 281-309)

(7) Lines 212-213: “The classification performance of the AM-CNN model is evaluated on two bird image datasets: The North American bird dataset (CUB200-2011) [21]”. Usage of North American bird dataset should be mentioned in the Methods section.

Answer: Good suggestion. In the revised manuscript, we added the new section 2.4 to introduce the training process of the AM-CNN model and the parameter settings, including all bird image datasets used (Lines:210-222).

(8) Line 287: The study contributes to citizen science, and this can be mentioned at the end of the Conclusion section with such a sentence or similar sentence that mention the contribution to citizen science “Our application can be used by anyone who have interest to water birds. In this context, our study contributes to volunteers for birdwatching/bird counting project and citizen science”

Answer: Good suggestion and thanks. In the revised Conclusion section, we have re-summarized the concluding contents, taking into account the suggestions of the reviewers.

Author Response

Dear reviewers,

Thank you for your letter and comments concerning our manuscript entitled “Eyebirds: Enabling the Public to Recognize Water birds at Hand” (Manuscript ID: animals-1970104). Your comments have proved very helpful with regard to our revisions and improvements of our paper, as well as providing insightful guidance for our research. We have gone through your comments carefully and have made necessary corrections. We have tried our best to improve the manuscript and we have made many changes to the text according to the comments of reviewers. All changes have been highlighted in red in the revised manuscript.

We sincerely appreciate the effort and consideration of reviewers and we hope that the corrections will meet with your approval. Once again, we thank you very much for your insightful comments and suggestions.

Here, we attached a summary of our major changes in the revision.

(1) We have revised unclear statements and some inaccurate words to make the manuscript more readable and concise.

(2) Based on the reviewers' suggestions, we have revised some inaccurate or redundant statements and added some important references.

(3) Referring to the reviewer's suggestion, we have added a description of the app's operation mechanism and application cases in section 2.3.

(4) Following the reviewer's suggestion, we have added a new section 2.4 to provide additional information on the experimental configuration and the model performance evaluation metrics.

(5) In light of the reviewers' suggestions, we have added a new section 3.4 to discuss the strengths and weaknesses of the model performance.

(6) Finally, we have reorganized the conclusions by combining the reviewers' suggestions.

Our point-to-point response to the comments is as follows:

(1) The authors state that recording bird sound requires specialized equipment and that the results “are usually of low quality… This makes it difficult to widely apply bird sound-based classification techniques to help the public recognize birds”. This is not accurate.

Answer: Thank you for pointing out our mistake and reminding us of the BirdNET model. In the revised version, we have corrected the original error and added relevant reference (Lines:37-38).

(2) Because this is a mobile app, I assume the authors imagine users uploading photos they have taken on mobile devices (otherwise, it might as well be a desktop app). However, it can be extremely challenging to take a decent quality photo of water birds with a cell phone. Figure 3 shows an example of the interface. Because I cannot read Chinese, I do not know whether all the images provided are examples of the what the app could provide after an identification has been made, or whether one or more are being used as examples of an image that might be uploaded for identification. I hope they are examples of what the app shows the user after identification, because those pictures would require thousands of dollars of camera equipment to capture and someone who has spent that much money on camera gear and takes the time to create such pictures can almost certainly identify the bird themself. In other words, a casual app user could never take those photos with a smart phone. This raises several questions: first, how exactly will the app work? Second, is the training data representative of the real-world input data the AM-CNN would be classifying? The low-quality images that the authors filtered out of their training data could actually be valuable as they represent real-world examples of the type of data the AM-CNN would be working with.

Answer: Very insightful questions and thanks.

In fact, after we developed the AM-CNN model in 2020, there was a debate in our team about whether to develop a mobile app. The manuscript's co-author, Dr. Yeai Zou, an experienced ornithologist, argued that smartphones do not provide high-quality bird photos and that professional birdwatchers or bird biologists already know bird species and do not need an app. However, most of our team insisted that a mobile app would be valuable to help the public identify birds and gain knowledge about birds.

To confirm whether the public is interested in our mobile app or not, we went to wetland parks, zoos, and wetland bird watching sites to conduct questionnaires. We asked visitors 2 main questions: (1) Do you want a mobile app that can help you identify birds and gain knowledge about birds? (2) If you like to use a mobile app to get bird knowledge, what features do you think the app should provide?

We found some interesting results: (1) Most of the public at these sites used smartphones to take photos of birds of interest, rather than professional cameras. (2) The general public, especially most parents with children, were willing to use a mobile app to inform their children about birds of interest. (3) Most professional birdwatchers with expensive cameras were not interested in mobile apps, but yet argued that mobile apps were helpful in recognizing birds, especially in the early stages of their birdwatching activities. (4) The public who are willing to use apps want apps to provide bird image identification and bird information, while professional birdwatchers want to have apps that record information about wild bird surveys. As can be seen, a bird image recognition app serves more the general public than professional birdwatchers or bird biologists. After all, to a certain extent, we live in the era of smartphone control.

We're sorry that the language version of the app made it difficult for you to read. For financial reasons, we have only launched the Chinese version for now, but we will provide the English version in the future. In addition, we also further explained the functions of the app and application examples. In the revised manuscript, we have also made relevant additional explanations (Lines:1995-199, 206-209).

The EyeBirds app is easy and convenient to operate. It mainly provides three functions of bird image recognition, introduction of bird information, and bird field survey. Under the condition of a smartphone network connection, users scan the QR code of WeChat to open the app directly. When users upload the images of birds of interest to the system, EyeBirds system receives and processes user task requests, and finally returns bird information about the top 3 bird species in terms of matching confidence. Usually the higher the probability, the higher the accuracy. Of course, a high probability value of a prediction result does not mean that the prediction is correct. To reduce misinformation to users, for each of the top 3 prediction results, the model provides a high-resolution bird reference image and bird information.

In fact, Figures 3(B and C) and Figure S1 reflect an example where a user uploads a bird image of interest (for instance, white-throated kingfisher), and the app returns the three highest matching bird species and related bird information. Users can refer to the bird image and text information provided by the model to identify whether the model prediction results are true or false (Figures 3- C and Figure S1). In addition, users can also use the app's bird field survey function to record the geographic coordinates of bird sites found, as well as the names of bird species and the number of bird species (Figure 3-D).

Of course, AM-CNN is essentially a deep learning model that has an inherent lack of current AI techniques: namely, its performance relies on a large inputs of training data with reliable quality. Therefore, we try to use good quality bird images when training the AM-CNN model. Only a small amount of our self-constructed water bird images was taken by volunteers via smartphones. However, considering the rapid updating of smartphones, it is to some extent possible for smartphones to take bird image photos of high quality in the near future. For example, one of the latest smartphone (Huawei mate40 pro) can be capable of up to 50x digital zoom, which actually hints the potential feasibility of replacing cameras with smartphones in the future.

  ( Please see Figure 3 and Figure S1 in the attachment, the figure cannot be displayed here)

(3) How was model performance measured (line 229)? False positive rate? false negative rate? Ratio of false positives to true positives? More detail would be valuable. I am guessing it is: correct predictions/total predictions. Is the algorithm constrained to always make a prediction (i.e., only possible outcomes are correct prediction and incorrect prediction) such that false negatives do not occur?

Answer: Good suggestion and thanks.

In our manuscript, model performance is really characterized by a classification accuracy metric. Classification accuracy is the ratio of the number of correct predictions in the testing sample to the number of testing samples. In the revised version, we explained this issue in a new section 2.4 (Lines:210-222).

  However, when users use the app to identify bird images, the model outputs the top 3 results with the highest probability. Usually the higher the probability, the higher the accuracy. Of course, a high probability value of a prediction result does not mean that the prediction is correct. To reduce misinformation to users, for each of the top 3 prediction results, the model provides a high-resolution bird image and bird information (Figure 3 and Figure S1). Users can refer to the bird image and text information provided by the model to identify whether the model prediction results are true or false.

 (4) The conclusions section seemed extremely short. How many people are using the app? Over time, how might the authors test their assumption that app usage leads to positive attitudes towards water bird and wetland conservation?

Answer: Good questions.

Our app has been free for public use since last September. The app has mainly spread to the public in Huai'an area through volunteers, and the number of users is not high, about 10,000. If the app is to be used by the public in a wider area, a larger platform and dedicated funding will be needed to promote it. This is something that our team lacks at the moment. However, we are doing our best to get funding support from different sources, even though it is difficult to do so in China.

Whether the app usage leads to positive attitudes of the public towards water bird and wetland conservation or not is really important. In fact, our team is preparing to implement a 3-year user tracking program next year, and we will obtain feedback data from users in order to evaluate the changes of their attitudes towards bird conservation. We will have the answer 3 years from now.

(5) 29: genetics is also an essential part of scientific classification

Answer: Good suggestion. Considering the other reviewer’s comment that the sentence of Lines 27-29 makes no sense, we replaced it with a new sentence “It is import to know or identify the taxonomic levels of birds for conservation efforts.” We de-emphasized the principle of bird classification.

(6) 39: The BirdNET algorithm is another important example.

Answer: In the revised version, we have corrected the original error and added relevant reference (Lines:37-38).

(7) carapace is an exoskeleton segment. birds do not have an exoskeleton. this term is used elsewhere (eg, line 96), so the problem is potentially extensive. similarly, trunk is not generally used to describe bird morphology.

Answer: Good suggestions.

In the previous manuscript, we used inaccurate words. In the revised draft, we have replaced both “carapace” and “trunk” with “torso”.